# Progress in Microbial Fertilizer Regulation of Crop Growth and Soil Remediation Research

**DOI:** 10.3390/plants13030346

**Published:** 2024-01-24

**Authors:** Tingting Wang, Jiaxin Xu, Jian Chen, Peng Liu, Xin Hou, Long Yang, Li Zhang

**Affiliations:** 1College of Plant Protection, Shandong Agricultural University, Tai’an 271002, China; 18653883150@163.com (T.W.); 17866710916@163.com (J.X.); liupeng2003@sdau.edu.cn (P.L.); houxin@sdau.edu.cn (X.H.); 2Institute of Food Quality and Safety, Jiangsu Academy of Agricultural Sciences, Nanjing 221122, China; chenjian@jaas.ac.cn

**Keywords:** microbiological fertilizer, plant-growth-promoting bacteria, crop growth, soil remediation

## Abstract

More food is needed to meet the demand of the global population, which is growing continuously. Chemical fertilizers have been used for a long time to increase crop yields, and may have negative effect on human health and the agricultural environment. In order to make ongoing agricultural development more sustainable, the use of chemical fertilizers will likely have to be reduced. Microbial fertilizer is a kind of nutrient-rich and environmentally friendly biological fertilizer made from plant growth-promoting bacteria (PGPR). Microbial fertilizers can regulate soil nutrient dynamics and promote soil nutrient cycling by improving soil microbial community changes. This process helps restore the soil ecosystem, which in turn promotes nutrient uptake, regulates crop growth, and enhances crop resistance to biotic and abiotic stresses. This paper reviews the classification of microbial fertilizers and their function in regulating crop growth, nitrogen fixation, phosphorus, potassium solubilization, and the production of phytohormones. We also summarize the role of PGPR in helping crops against biotic and abiotic stresses. Finally, we discuss the function and the mechanism of applying microbial fertilizers in soil remediation. This review helps us understand the research progress of microbial fertilizer and provides new perspectives regarding the future development of microbial agent in sustainable agriculture.

## 1. Introduction

The global population will continue to grow and is expected to exceed 9 billion by 2050, requiring a rapid increase in crop production [1]. Urbanization and industrialization have led to a significant reduction in arable land while at the same time causing damage to agroecosystems. The need to increase crop yields to feed a growing population on limited arable land requires significant inputs of agrochemicals into the agricultural environment [2]. However, the long-term use of these agrochemicals not only affects soil fertility but also contaminates agroecosystems through the introduction of toxic substances. These pollutants are persistent and accumulate in the environment over time, leading to the further contamination of the food chain and posing a threat to human health [3,4]. This limits the sustainable productivity of soils and poses a threat to the environment and human health. The fact that microorganisms are an important natural resource for the development of “green methods” has attracted great attention worldwide. Improving soil health and crop growth through the use of fertilizers enriched with beneficial microorganisms is essential to maintaining a balance between efficient crop production and sustainable agriculture [5].

The inter-root is an active area for microbial interactions with the plant root system [6]. In inter-root soils, plant growth benefits from several PGPR with multiple functions, and these beneficial bacteria play important roles in soil nutrient cycling, the decomposition of organic matter, the suppression of soil-borne diseases, and the improvement of crop growth and resistance. Some examples are biological nitrogen fixation, phosphate solubilization, phosphorus and potassium solubilization, hormone secretion, the inhibition of pathogens, the induction of plant resistance, etc. [7,8]. In addition, plants also secrete secondary metabolites to feed the microorganisms around the inter-root. Interactions between plants and beneficial bacteria play an important role in maintaining the microenvironment around the plant root system, which is favorable for plant growth and development [9].

Microbial fertilizer is a type of bio-fertilizer containing beneficial microorganisms with specific functions [10]. The application of microbial fertilizers can help the soil form a new microbial community structure and promote the supply of nutrients to plants with the soil [11]. Numerous studies have shown that microbial fertilizers made from PGPR play an important role in improving soil fertility and promoting crop growth. They also help crops combat biotic and abiotic stresses. Today, microbial fertilizers are considered renewable and environmentally friendly, supporting sustainable agriculture. In this review, we summarize the mechanism of action of microbial fertilizers in promoting crop growth and resistance to environmental stresses, and we also discuss the application of microbial fertilizers in soil remediation.

## 2. Microbiological Fertilizer Classification

Microbial fertilizers can be classified into different types according to their functions. Briefly, they can be divided into three categories: bio-organic fertilizer, compound microbial fertilizer and microbial fungicide. Table 1 summarizes the beneficial growth-promoting bacterial strains of different types of microbial fertilizers on different crops.

### 2.1. Bio-Organic Fertilizer

Bio-organic fertilizer is primarily composed of livestock and poultry manure, crop residues, municipal waste, and other organic waste materials that have undergone harmless treatment. It is enriched with specific functional microorganisms and serves as a compound fertilizer for organic matter. Bio-organic fertilizer offers the dual benefits of microbial fungicide and traditional organic fertilizer. Along with its high organic matter content, it also contains specific microorganisms with unique functions [12]. In recent years, the rise in organic waste and pollutants has resulted in a gradual decline in farming quality, which subsequently impacts soil health [11]. Through the formulation of fertilizer application and the return of straw to the field and other measures, the fertilizer effect of organic fertilizer can be brought into play and the threat to the soil environment posed by excessive chemical fertilizer can be reduced, which is conducive to the promotion of the development of bio-organic fertilizer [13]. Compared with chemical fertilizer, bio-organic fertilizer has the efficacy of traditional organic fertilizer on the one hand and the special function of beneficial micro-organisms on the other hand; through the combination of the two, it plays the role of maximum fertilizer efficiency [14]. It has been found that bio-organic fertilizers often act as beneficial soil conditioners, improving soil quality, promoting the formation of soil aggregates, and enhancing the availability of nutrients to crops [15]. It has also been shown that bio-organic fertilizers increase soil microbiota and promote soil carbon and nitrogen cycling compared to chemical fertilizers. The most important function of bio-organic fertilizer is to recruit more beneficial soil microorganisms through the beneficial microorganisms it contains and colonize the inter-root area in large quantities, which is the place where microorganisms, soil, and plant roots interact with each other to allow better communication between them.

### 2.2. Microbial Agents

Microbial agents, also known as microbial inoculants, are defined as biofertilizers containing a range of live microbial products [16]. Beneficial bacteria with specific functions are isolated from bacteria, fungi, actinomycetes, and algae to develop microbial inoculants [17,18]. Currently, new inoculants represented by PGPR have attracted widespread attention. PGPR, such as *Klebsiella*, *Azotobacter*, *Azoospirillum*, and *Bacillus*, proliferate in the inter-root soil. *Bacillus* and *Pseudomonas*, which are isolated from the rhizosphere, have been extensively utilized for the development of microbial inoculants based on the identification of microorganism diversity [19]. Research demonstrates that microbial inoculants incorporate a specific range of beneficial bacteria, which induce hormonal substances for plant growth through their inherent direct mechanisms, and counteract the threat of biotic or abiotic stress cues to plants. These inoculants do not pose a pathogenic effect on plant organisms, are environmentally friendly, promote plant adaption to the environment, enhance growth and development, harmonize nutrients in soil, yield diverse plant growth regulators, detoxify soil heavy metals, pesticides, and fungicides, and utilize biological control techniques for soil remediation. The initial studies on microbial inoculants primarily involved single strains that had singular functions and minimal impact on plant growth [20]. However, current research on microbial inoculants has shifted toward investigating multiple beneficial bacteria and multifunctional composite inoculants. As a result, this area of study has gained significant attention.

### 2.3. Complex Microbial Fertilizer

Composite microbial fertilizer integrates a variety of beneficial bacteria such as *Bacillus subtilis*, *Bacillus licheniformis*, *Azospirillum brasilense*, and *Streptomyces*, synergistically activating characteristics such as the solubilization of phosphorus and potassium, and nitrogen fixation through optimal combination [21,22]. Compound microbial fertilizer primarily involves agricultural residues (avian excreta and straw) and beneficial microorganisms as the primary raw materials to produce a novel variety of fertilizers [23,24]. Complex microbial fertilizers incorporate chemical fertilizers, organic fertilizers, and beneficial microorganisms. They foster robust plant growth while possessing the immediacy of chemical fertilizers and the longevity of organic fertilizers. This effectively improves soil fertility and sustains healthy, thriving crops [25]. In this case, the beneficial bacteria contained in the composite microbial fertilizer produce secondary metabolites through the metabolic activities of the microorganisms, which dissolve the mineral elements in the soil and promote crop growth. Root-associated beneficial microorganisms are colonized, establishing a beneficial microbiome. This further interacts with rhizosphere secretions from the beneficial microorganisms to induce secondary metabolite production in plants, participate in plants’ defense systems, and produce growth regulators that promote plant growth and regulate crop development.

**Table 1 plants-13-00346-t001:** Beneficial strains of growth-promoting bacteria of different types of microbial fertilizers in different crops.

	Types of Microbial Fertilizers	Crop	Plant Growth Promoting Rhizobacteria	References
1	Bio-organic fertilizer	Lettuce	*Actinobacteria*, *Proteobacteria*, *Chloroflexi*, *Acidobacteria*, *Gemmatimonadota* *Ascomycota*, and *Basidiomycota*	[26]
2	Bio-organic fertilizer	Tobacco	*Actinobacteria*, *Chloroflexi*, *Proteobacteria*, *Acidobacteria*, *Firmicutes*, *Gemmatimonadota*, *StreptomyceBacillus*, *Arthrobacter*, and *Paenibacillus*	[27]
3	Bio-organic fertilizer	Beet, potato, winter, wheat	*Actinobacteria*, *Proteobacteria*, *Acidobacteria*, *Arthrobacter*, and *Paenibacillus*	[28,29]
4	Bio-organic fertilizer	Cauliflower	*Proteobacteria*, *Actinobacteria*, *Acidobacteria*, *Gemmatimonadetes*, *Bacteroidetes*, and *Chloroflexi*	[30]
5	Bio-organic fertilizer	Tomato	*Proteobacteria*, *Actinobacteriota*, *Bacteroidota*, *Firmicutes*, *Firmicutes*, and *Verrucomicrobiota*	[31]
6	Microbial inoculants	Watermelon	*Pseudomonas*, *flavobacterium* *Aspergillus*, *Myceliophthora*, *Trichoderma*, and *Humicola* and *Neocosmospora*	[32]
7	Microbial inoculants	Radish	*Proteobacteria*, *Bacterioidetes*, *Acidobacteria*, *Actinobacteria*, and *Planctomycetes*	[33]
8	Microbial inoculants	Rice	*Proteobacteria*, *Acidobacteria*, *Bacteroidetes*, *Gemmatimonadetes*, *Actinobacteria*, *Planctomycetes* *Ascomycota*, and *Chytridiomycota*	[34]
9	Microbial inoculants	Prunusdavidana	*Proteobacteria*, *Bacteroidetes*, *Acidobacteria*, *Gemmatimonadetes*, *Actinobacteria*, *Patescibacteria*, *Chloroflexi*, *Verrucomicrobia*, *Nitrospirae*, *Latescibacteria*, and *Rokubacteria*	[35]
10	Microbial inoculants	Cucumber	*Alphaproteobacteria*, *Actinobacteria*, *Acidobacteria*, *Betaproteobacteria*, *Gammaproteobacteria*, *Deltaproteobacteria*, *Gemmatimonadetes*, *Bacteroidetes*, *Chloroflexi*, *Planctomycetes*, *Firmicutes*, *Verrucomicrobia*, *Nitrospirae*, *Armatimonadetes*, *Cyanobacteria*, *TM7*, *Fibrobacteres*, and *Chlorobi*	[36]
11	Compound microbial fertilizer	Soybean	*nitrogen-fixing bacteria*, *phosphorus-solubilizing bacteria*	[37]
12	Compound microbial fertilizer	Sugarcane	*Trichoderma harzianum*, *Gluconcetobacter diazotrophicus*, and *Pseudomonas fluorescents*	[38]

## 3. Microbial Fertilizers Regulate Crop Growth and Resistance

### 3.1. Microbial Fertilizers Regulate Crop Growth

Plant growth and development are intricately linked through mutual interactions between organisms in the root of plants and the rhizosphere. The rhizosphere prevails as a distinct habitat for the majority of microorganisms, serving as a direct source of their nutrients. In particular, bacteria that live around the roots, for example PGPR, are recognized as beneficial bacteria that promote plant growth and regulate plant growth and development. PGPR regulate plant growth and enhance yield via diverse direct action mechanisms, further augmenting plant nutrient assimilation [39]. The mechanisms of the direct action of PGPR encompass nitrogen fixation, the solubilization of phosphorus and potassium minerals, the generation of plant hormones, and the production of ferric iron carriers [40]. Table 2 illustrates the mechanisms by which various beneficial microorganisms regulate crop growth.

#### 3.1.1. Nitrogen Fixation

Nitrogen is a critical nutrient element in the growth and development of plants. As the nitrogen in the atmosphere exists in its free state, most plants are incapable of directly utilizing it. Thus, the necessity arises to fix the free nitrogen and transmute it into nitrogen that can be absorbed and utilized by plants. Significantly, a plethora of studies have demonstrated that nitrogen can be fastened in an assimilable condition for crops via a distinct group of microorganisms, which are designated as biological nitrogen fixers (BNFs). These elements facilitate the colonization and nitrogen fixation of rhizosphere bacteria [55].

Nitrogen-fixing bacteria are non-symbiotic, free-living bacteria. They belongs to the family of *Azotobacteria* and are mainly used as a non-leguminous crop biofertilizer. The main nitrogen-fixing bacteria include symbiotic nitrogen-fixing bacteria, free-living nitrogen-fixing bacteria, and combined nitrogen-fixing bacteria, and the nitrogen-fixing bacteria associated with leguminous plants include *Rhizobium*, *Azotobacteria*, and slow-growing rhizobacteria [56,57], Associative nitrogen-fixing bacteria with nonleguminous plants encompass *Arthrobacter*, *Alcaligenes*, *Mycobacterium*, *Pseudomonas*, *Bacillus*, and *Azospirillum* [58]. Among them, *Rhizobium* represents the epitome of symbiotic nitrogen fixation and is one of the most comprehensive studies on the symbiotic relationship between root nodules and nitrogen-fixing bacteria [59]. *Rhizobium* perceives flavonoid compounds exuded from plant root systems to produce signals for reciprocal communication with the root microbiome [60]. For instance, secretions from peanut root can stimulate the rhizobium of peanuts. *Rhizobium* is attracted to the root’s secretions, then forms nodules and fixes nitrogen in the roots of the host plant [61]. In addition, it was observed that the association of alfalfa with *Rhizobium* can enhance biological nitrogen fixation, stimulate plant growth, and more pertinently, rehabilitate heavily metal-contaminated soil while augmenting the resistance of plants to metals [62].

#### 3.1.2. Phosphate Solubilizing

As an indispensable second nutrient element for plant growth and development, phosphorus contributes to important physiological processes such as plant metabolism, root growth and development, and flowering and fruiting [63]. Due to long-term application of chemical fertilizers, more than 70% of phosphorus in soil exists in inorganic form, and this inorganic phosphorus can easily react with Fe^3+^, Al^3+^ and Ca^2+^ in soil to form insoluble phosphate [64]. Consequently, the addition of beneficial microorganisms is required to solubilize phosphates from the soil. It has been reported that species of bacteria and fungi, including *Bacillus*, *Rhizobium*, *Pseudomonas*, *Penicillium*, *Aspergillus*, and *Staphylococcus*, are typical phosphorus-enhancing agents [65]. The presence of insoluble phosphate in soil requires its conversion into soluble phosphate by soil microorganisms. Soil bacteria and fungi are involved in the solubilization process of soil phosphate by producing different mechanisms of action. Some of the mechanisms of action include the secretion of organic acids by beneficial bacteria, the formation of chelates, ion exchange reactions, etc. [66]. These beneficial microorganisms secrete organic acids that lower soil pH, thereby increasing the effectiveness of phosphorus in the soil [67]. Through these dissolved phosphorus microorganisms producing organic acids, secreted enzymes, iron carriers, etc., metal ions in the soil are chelated to form a complex, which is converted into phosphate, which can be absorbed and utilized by plants [68,69]. They contribute to phosphate solubilization, increase phosphate utilization by the plant, and enhance physiological processes in the plant.

#### 3.1.3. Potassium Dissolution

Potassium is the third most essential nutrient, following nitrogen and phosphorus. Potassium is abundant in soil and exists in various forms. It plays a pivotal role in a plant’s growth and development, influencing plant growth, root system development, and enhancing yield and quality. The majority of potassium exists as in mineral form, which plants cannot directly assimilate. It has been documented that utilizing the distinct mechanisms of bacteria and fungi enables the transformation of insoluble potassium into soluble potassium. For instance, *Bacillus*, *Arthrobacter*, *Azotobacter*, and *Aspergillus* are archetypal potassium solubilizers. Research indicates that the *Bacillus and Klebsiella* isolated from the root zone of chili could solubilize substantial amounts of potassium in the soil, with the potential for potassium dissolution to exceed 70% compared to that of the control [70]. Recently, a study has indicated that the isolation of potassium-solubilizing bacteria from the rice rhizosphere enhances soil potassium availability and subsequently stimulates growth and yield in rice [71].

#### 3.1.4. Regulating Phytohormone Levels

Additionally, phytohormones, also referred to as plant stimulators, are produced by PGPR and promote plant development [49]. The growth and development of crops are controlled by stimulating the plants. Microorganisms manufacture phytohormone, which are organic substances that regulate various aspects of plant growth, including cell division and differentiation, organ development, fruit ripening, flower blossoming, and fruiting. Auxins, cytokinins, gibberellins, ethylene, and abscisic acid are examples of phytohormones. Table 3 summarizes examples of phytohormones produced by different microorganisms in crops.

The widespread plant growth regulator auxin (also known as indole-3-acetic acid, indole-acetic acid, or IAA) is essential for promoting vegetative development [72]. The majority of PGPR generate indole acetic acid, which is essential for coordinating the host plant’s and microbiota’s interaction. In the rhizosphere, tryptophan (Trp), a substance found in root exudates, is exchanged between the plant-beneficial rhizobacterium and the host plant to facilitate communication [73]. As a signaling molecule that facilitates communication between Rhizobia and the host plant, tryptophan triggers PGPR to synthesize IAA through a variety of pathways [74]. IAA is an intrinsic regulator of plant development that mainly controls the growth of plant roots, promotes the creation of root hair, and stimulates the genesis of root epidermal cells. Furthermore, IAA influences photosynthesis, promotes plant cell differentiation and division, and aids in the creation of vascular bundles. One study used Helicobacter *Azotrophicus* inoculation to examine the impacts of root development in Arabidopsis thaliana in Brazil. The findings indicated that the growth hormone encouraged the production of lateral root meristematic tissues [75].

An essential plant hormone required for plant growth and development is cytokinin. It is primarily produced by inter-root bacteria, which use the synthesis of cytokinins to stimulate and support plant growth [76]. Plant growth and development activities, including root development and hair creation, stem and root elongation, light response regulation, and stomatal open-in promotion, are primarily driven by cytokinins. Furthermore, research has demonstrated that gibberellins (GA) are also produced by PGPR. Gibberellins primarily break dormancy in seeds and encourage germination; they also lengthen stems, induce the growth of floral organs, and increase fruit set. For example, gibberellin can protect the host plant against stress-related dangers when environmental conditions are unfavorable. Furthermore, certain species of inter-root bacteria release and manufacture ethylene, a special regulator of plant growth. They primarily control the growth and development of plants, encourage the growth of roots, and quicken the ripening of fruit. Ethylene’s mode of action greatly increases plant vegetative growth and development, hastens plant maturity, which lowers plant consumption of soil nutrients, and increases the amount of soil nutrients that can be retained in the soil, minimizing the need for phosphorus and potassium fertilizers.

#### 3.1.5. Iron Carrier Production

Iron is essential for the growth and metabolism of plants, primarily for the regulation of several physiological processes. For instance, respiration, photosynthesis, and nitrogen fixation all guarantee the availability of nutrients for plant growth. Because of its great susceptibility to oxidation, iron in soil can precipitate insoluble iron oxide, which plants cannot absorb [77]. The slow rate of decomposition of the majority of the inorganic iron minerals in rhizosphere soil inhibits the growth and development of plants. By employing a variety of mechanisms, PGPR can increase iron solubility by generating a number of tiny molecules known as iron carriers [78,79]. Low-molecular-weight, organic secondary metabolites generated by specific bacteria are called iron carriers [80]. These organic compounds are mostly used by microorganisms as iron chelators, helping them to absorb iron. These iron chelators can reduce the stress that heavy metals place on the environment by chelating iron as well as other heavy metals. Iron carrier complexes are created on the cell membrane by metabolites released by iron carriers interacting with Fe^3+^. Eventually, these complexes are broken down to Fe^2+^, which plants can absorb and use to support development. Numerous studies support the idea that *rhizobacteria* connected to plants that promote plant growth are also able to produce iron transporters. For example, by producing iron carriers and P solubilization, PGPR isolated from soil can improve the nutritional growth features of tomato plants, thus increasing nutrient efficacy [81]. Importantly, the *Burkholderia* P10 strain clarifies how the P10 transcriptome affects the transformation of peanut root exudates. Furthermore, it greatly amplifies the P10 strain’s promotion of plant development by promoting the biosynthesis of iron carriers, the synthesis of IAA, and the expression of genes linked to phosphorus dissolution [82].

**Table 3 plants-13-00346-t003:** Examples of phytohormones produced by different microorganisms in crops.

Microorganisms	Crop	Phytohormones	Mechanism of Action	References
*Rhizophila* Y1	Corn	IAA, ABA	*Rhizophila* Y1 regulates phytohormone levels and alleviates salt stress in maize growth	[83]
*Bacillus velezensis*	Strawberries	IAA	*Bacillus velezensis* produces large amounts of IAA for growth promotion	[84]
*Bacillus thuringiensis* RZ2MS9	Corn	IAA	Genetic basis for the induction of IAA biosynthesis by *Bacillus thuringiensis* RZ2MS9 for maize growth	[85]
*Leifsonia soli* SE134	Cucumbers, Tomato	GA	GA secretion by *L. soli* SE134 may favor its ameliorative role in crop growth	[86]
*Bacillus subtilis*	Maize, Brassica pekinensis	GA	*Bacillus subtilis* secretes gibberellins that promote the growth of rice and cabbage	[87]
*Bacillus subtilis*	Wheat	GA, IAA	*Bacillus* sp. increases endogenous IAA and GA levels in all genotypes of wheat	[88]
*B. subtilis* CNBG-PGPR-1	Tomato	Ethylene	*B. subtilis* CNBG-PGPR-1 regulates the ethylene pathway in tomato, scavenges ROS, and enhances plant salt tolerance	[89]
*Bacillus subtilis*	Corn	Ethylene	Salt-tolerant *Bacillus* sp. strains reduce stress-inducing ethylene levels in host plants and alleviate salt stress	[90]

### 3.2. Increasing Crop Resistance to Environmental Stress

Reactive oxygen species (ROS) produced during environmental stress during plant growth impair plant productivity by causing organelle damage and eventual cell death [91]. Plants experience both biotic and abiotic stressors during their reproductive processes, leading to lower agricultural yields [92]. Fungi, bacteria, viruses, nematodes, and other biological creatures are all included in biotic stress [93]. Fungi, bacteria, viruses, nematodes, and other biological organisms are all included in biotic stress [94]. To maintain the agricultural ecological balance, according to the pressure encountered, we should implement effective solutions to the current environmental pressure and deal with the influence of various pressures [95]. As a result, some defense mechanisms are used to lessen the severe stresses that plants face [96]. On one hand, the genetic modification of crop varieties can foster robust crops that can withstand environmental fluctuations. However, it is crucial to consider that this process of nurturing resistant crop varieties is time-consuming and demands a considerably high investment in time. Currently, the principal methodology employed to alleviate plant environmental stress is the utilization of advantageous growth-promoting microbe of root zone soil [97]. PGPR impart antagonistic substances in the rhizosphere through suppressing pathogen growth and competing for nutrients [98]. PGPR mainly use various metabolites and volatiles to regulate the structure of soil microbial communities, suppress soil pathogens, and improve soil health. For example, PGPR produce antibiotics, hydrolyte enzymes, and antimicrobial compounds for use in attacking pathogen growth, thereby protecting plants from pathogens. PGPR improve plant growth and resistance through a number of mechanisms of action, providing effective alternatives to traditional control methods [99].

#### 3.2.1. Biotic Stress

Biological Control of Pest Management

Previously utilized pesticides and insecticides for biological control have a discernible negative impact on soil and human health. Some plant growth-inducing bacteria can protect plants from pests through pathogenic mechanisms, metabolites, and secretions. However, PGPR-based biocontrol agents are effective alternatives to synthetic pesticides and insecticides. For example, *Bacillus* and *Pseudomonas* are effective against pests [100]. *Bacillus thuringiensis* (Bt) is a prominent plant growth-promoting rhizobacterial biological insecticide, extensively applied to noctuidae, coleoptera, and diptera in insect classification [101]. *Bacillus thuringiensis* is a bactericidal protein and a quick-acting insecticide possessing minimal side effects on host plants and other beneficial micro-organisms [102]. It has been reported that *Bacillus* 90 and *Pseudomonas aeruginosa* strain 91k were evaluated in wheat studies and found to be effective against aphid populations [103]. PGPR secretion of volatile organic compounds helps defend against nematode damage and triggers ISR to resist pathogen attack [93]. It has been reported that the use of PGPR is effective in controlling potato nematode damage without secondary environmental pollution to the environment [104].

Plant Pathogen Management

The defense mechanisms of PGPR protect against pathogenic bacteria, viruses, and fungi by inducing systemic resistance (ISR) and systemic acquired resistance (SAR) in plants [105,106]. PGPR-dominant strains that induce ISR include *Pseudomonas*, *Bacillus*, and *Serratia*. ISR is an activation response induced by a diverse array of beneficial and detrimental microorganisms, as well as environmental stressors. It is not possible to unequivocally discern the mechanism that induces ISR [106,107]. The primary mechanism by which PGPRs defend plants from biotic stress is their capacity to synthesize antibiotics [108]. Antibiotics are polyvalent microbe-derived substances of a low molecular weight possessing toxic organic components that can enhance plant growth and various metabolic activities [109]. Antibiotics are classified into two categories, namely volatile complex compounds and non-volatile complex compounds. Significantly, different types of beneficial bacterial genera in PGPR can produce antibiotics as a potent method to combat the invasion of pathogens. The dominant genus taxa producing antibiotics include *Pseudomonas fluorescens*, *Bacillus*, *Actinobacteria*, *Enterobacteriaceae*, and *Arthrobacter*. Antibiotics not only provide direct resistance to pathogens but also promote disease suppression in plant systems by inducing systemic resistance, conferring a competitive advantage to biocontrol agents. Nowadays, *Pseudomonas fluorescens* has the potential to protect against plant pathogen attacks as an effective control agent against plant pathogens [110]. *Bacillus* is a dominant genus within PGPR, efficiently combating plant pathogens. *Bacillus* also produces distinct antibiotic classes, primarily *Bacillomycin*, *Rhizobiumin*, and *Mycobacteriumin.* It is also known to generate antimicrobial surface-active agents [106]. In particular, *Bacillus* subtilis stops pathogens through biological control. In addition to this, *Bacillus* can produce iron carriers and extracellular polysaccharides to help regulate ionic balance and synthesize microbial metabolites to help control the threat of plant diseases. In addition, hydrolytic enzymes mainly include cellulases, proteases, chitinases, and lipases. Their main mechanism of action contributes to the hydrolysis of polymeric compounds, cleaving the cell walls, proteins, and DNA of pathogens to protect plants from pathogens. These volatile organic compounds can regulate the structure of soil microbial communities, in turn affecting the growth and development of fungi, plants, and animals. The accumulation of beneficial soil microorganisms can be stimulated via the application of biofertilizer with exogenous beneficial bacteria, in turn leading to the formation of beneficial flora against pathogens and ultimately to the recruitment of more disease-resistant microorganisms.

#### 3.2.2. Abiotic Stress

Drought Stress

Among abiotic stresses, drought is an important factor affecting agricultural production. Water scarcity affects plant physiological processes, water–nutrient relationships, and the normal metabolic activities of the plant body [111,112]. Currently, several strategies are necessary to overcome drought stress. The use of PGPR as inoculants alleviates water supply deficiencies and effectively promotes water utilization. The primary mechanism for the palliative effect of PGPR in drought is derived from the regulation of plant hormones, volatile compounds, and cell wall polysaccharides, which influence the normal growth of crops [113]. Through these mechanisms of action, they helps plants maintain survival under extreme drought conditions. The synthesis of phytohormones by *Pseudomonas*, *Bacillus* and *Rhizobium* isolated by PGPR is conducted to stimulate plant growth and overcome the stress of drought stress [114]. Under arid conditions, the introduction of beneficial microbial agents can generate extracellular polysaccharides (EPS), synthesize proline, and secrete phenolic compounds, and regulate plant growth and development to resist dehydration stress [115]. Salicylic acid (SA), primarily produced by microorganism-derived phenolic compounds, serves as a critical signaling molecule in arid conditions. It effectively activates antioxidant genes and underived metabolic gene products to manage plant growth and development [116]. ACC is a precursor of ethylene, and PGPR-produced ACC deaminase can degrade ACC levels, preventing an excessive increase in ethylene and thereby resisting abiotic stress [117]. It has been reported that PGPR strains produce 1-aminocyclopropane-1-carboxylic acid (ACC) deaminase, which protects tomato from the negative effects of drought stress and significantly enhances the drought tolerance of plants [118]. Another study also reported that under drought and salt stress conditions, three beneficial PGPR isolates increased the IAA content, decreased the ABA/ACC content and improved the photosynthetic efficiency of wheat, thereby increasing its tolerance to abiotic stresses [119]. Research suggests that interspecific hybrid corn employs the PGPR-based isolation of *Pseudomonas putida*, *Pseudomonas fluorescens*, and *Bacillus megaterium* under conditions of drought stress. In the case of *Pseudomonas* putida treatment, seedling germination vitality, fresh and dry weight, dry matter content, and grain yield exhibit superior outcomes [120]. The inoculation of wheat in potting soil under drought conditions using two novel PGPR isolates, *Bacillus subtilis*-FAB1 and *Pseudomonas aeruginosa*-FAP3, stimulated plant growth and effective inter-root colonization, resulting in normal plant growth [121]. In recent years, the latest mechanisms of drought resistance have been mainly based on molecular and histological techniques to study some drought-resistant genes, contributing to our understanding of the multiple functions of rhizobia under drought conditions. It was shown that, using a metabolomics approach with untargeted liquid chromatography using UHPLC, sorghum was inoculated with key molecules for PGPR-induced tolerance to drought stress in sorghum [122]. Similarly, the characterization of chickpea physiological and biochemical traits under drought conditions via 16S-rRNA gene sequencing and the identification of the dominant PGPR strains such as *Bacillus subtilis* and *Bacillus thuringiensis* led to changes in the metabolome to reduce the effects of stress [123].

Salt Stress

In recent years, salt stress has become an important factor limiting plant growth. Excessive salinity leads to soil crusting, which further reduces the effective use of water by plants. Salinity directly affects the growth of the plant root system, which further impacts the entire growth process and metabolic activities of the plant [124]. Salinity directly affects chlorophyll content and carotenoids, denaturing the ultrastructure of the chloroplast, thereby reducing stomatal conductance and curtailing leaf photosynthesis. Increased levels of reactive oxygen species in plant cells are due to salt accumulation in the soil, leading to oxidative stress in plants [125]. Salinity can induce the accumulation of Na^+^ and Cl^-^, and at the same time impair the absorption of K^+^ and Ca^2+^, leading to an imbalance in ion homeostasis [126]. Consequently, it is necessary to employ strategic measures to mitigate the effects of salinity stress on plants. Extensive research indicates that PGPR can be leveraged to reduce crop yield losses resulting from salinity. PGPR influences plant physiological and biochemical processes through diverse mechanisms, mitigating the restrictions caused by salt stress on plant growth. Its primary mechanisms encompass the regulation of ion homeostasis, synthesis of protective agents against osmotic stress, activation of antioxidant enzymes, etc., all of which contribute to crop development [127]. Through the interplay of mechanisms and root zone microbiology, intricate signal networks regulate defensive mechanisms, alleviating stress [128]. It has been shown that inoculating rice seedlings with *Pseudomonas aeruginosa* and *Klebsiella* significantly increased plant height, root length, and plant dry weight, as well as promoting rice growth under salt stress conditions [129]. It has been reported that using *Bacillus* to investigate the growth of tomatoes under salt stress was found to induce systemic tolerance in tomato plants and had a significant impact on the diversity of the bacterial community [130]. It has also been reported that the beneficial *Bacillus* sphaericus SQR9 secretes spermidine, which induces salt tolerance in Arabidopsis thaliana and maize, enhancing their salt tolerance [131]. Therefore, PGPR can effectively resist the negative effects of salt stress, improve the salt tolerance of plants, and induce the development of resistance systems in plants.

Heavy Metal Stress

In addition to drought and salt stress, heavy metal pollution also has a negative impact on sustainable agricultural development. Due to increased heavy metal concentrations attributed to various anthropogenic activities, soil health has been compromised, directly influencing plant enzyme activity and nutrient transformation. Consequently, plant growth and development are hindered [132]. Thus, it is imperative to implement certain strategies to remediate contaminated soil. The utilization of microorganisms, specifically PGPR, in support of bioremediation technology has garnered widespread attention [133]. The primary objective of the microbiological remediation of soil heavy metals is to first immobilize the heavy metal, subsequently reduce its mobility, and ultimately remove it from the soil. The primary heavy metals include manganese, cadmium, iron, and zinc; the majority of microorganisms can absorb soil heavy metals through various mechanisms. The deposition in the affected soil exerts a detrimental effect on plant growth, root system development, photochemical properties, and nutrient assimilation [134]. The mechanism predominantly involves the binding of certain surface heavy metals by live microbial cells and surface-active substances. Concurrently, microorganisms undergo growth and metabolic processes that generate certain inorganic salts and hydrogen peroxide metabolites. These substances react with heavy metal ions to form precipitates [135,136]. In heavy metal-contaminated environments, microorganisms produce iron carrier chelators that can bind to various heavy metals, reducing their toxicity and promoting the growth of barley [137]. In a study on spinach, *Bacillus subtilis* and *Pseudomonas aeruginosa* were inoculated to enhance transpiration rate, stomatal conductance, and relative water content, as well as to improve resistance to heavy metal stress. This process also increased the capabilities of the antioxidant defense system [138]. Husna et al. found that rhizobial *Rhizobium* bioinoculants, which solubilize phosphate and release glycosides, helped alleviate metal stress in soybean seedlings, enabling them to better cope with chromium and arsenic toxicity [139].

In Table 4, we summarize the mechanism of beneficial microorganisms to alleviate the abiotic stress of crops.

## 4. Soil Remediation

Soil is the culmination of enduring formation processes, a virtually non-renewable resource and an indispensable element of the environment. It serves as a filter and reservoir for water, provides water and nutrients for plants to grow, and provides a habitat for a large number of organisms. Due to the relentless pursuit of crop yield, the repeated application of chemical fertilizers, pesticides, and toxic substances has led to their accumulation in the soil, culminating in environmental degradation globally. Healthy soil has been irrevocably lost. Revitalizing the health of soil and addressing the environmental issues impacting current soil conditions is a significant endeavor.

In the past, conventional restoration strategies encompassed physical and chemical methodologies; however, these techniques presented limitations such as high costs, lengthy temporal durations, and potentially induced secondary soil pollution. The paramount task at present is to identify methodologies for the sustainable restoration of the ecological environment. In addition to traditional approaches, microbial remediation has emerged as a contemporary, effective, and sustainable instrument for rejuvenating soil health [146]. Plants and microorganisms are usually utilized in the remediation of contaminated soil. Plants primarily utilize their inherent capacity to absorb contaminants, while microorganisms predominantly degrade them. Bioremediation technology primarily relies on microorganisms and is considered a sustainable and environmentally friendly approach to degrading environmental pollutants [147]. The soil microbiome, which is a crucial element of bioremediation, plays a significant role in the mechanisms of soil microbial remediation [148]. The essence of bioremediation technology lies in harnessing the capability of microorganisms to degrade metabolites and facilitate the transformation of pollutants in the soil [149]. Presently, within agricultural practices cognizant of biotechnological reparation, the recuperation of soil fertility via microbial fertilizers composed of diverse beneficial microbial strains is implemented. The interplay between plants, microorganisms, and soil is considered in rectifying soil pollutants thereby facilitating healthy plant growth by providing an advantageous environment.

Microorganisms employ diverse mechanisms to degrade toxic pollutants and pesticide residues within the soil. Currently, biofertilizers containing plant growth-promoting rhizobia represent crucial tools for remediating soil degradation [150]. The microbial remediation strategy will evolve into a vital soil management methodology. PGPR exhibit a positive influence on the soil itself, recruiting a larger number of beneficial microorganisms to effectively decompose the accumulation of toxic substances in soil [151]. They have the capacity to convert toxic organic compounds into nontoxic forms. The degradation of organic pollutants by microorganisms mainly occurs through the enzymes they break down, which help to catalyze the transformation of soil pollutants [152]. In addition, several hydrolytic enzymes can degrade the toxicity of toxic molecules and act as biodegraders [153]. The utilization of the symbiotic relationship among plants, microorganisms, and soil can be employed to remediate pollutants in soil and promote healthy plant growth. The process of decontaminating soil heavy metals used by PGPR involves various methods, including iron carrier chelation, biological adsorption, and biodegradation [154]. The bioremediation of soil heavy metals primarily occurs through the interaction of inter-root microorganisms with soil physicochemical properties, which in turn regulates plant growth and stimulates the detoxification of heavy metals in the soil [155]. Soil microorganisms secrete metabolites that can release iron carriers and organic acids, which aid in the chelation of soil toxic metal ions and contribute to the adsorption of heavy metals [156]. The EPS produced by PGPR can adsorb some metal ions such as Cu^2+^, Pb^2+^Cr^6+^, etc., and at the same time, it can degrade polycyclic aromatic hydrocarbons (PAHs) and alkane compounds, thus degrading pollutants in the soil [157]. Exploiting the beneficial rhizosphere microorganisms in soil to mitigate heavy metal stress, these organisms can accumulate, transform, and decompose the soil heavy metals [158]. Furthermore, the organic acids produced by microorganisms react with soil metal ions to dissolve heavy metal ions [159]. Previously conducted studies have indicated that microbial inoculation, including PGPR, has a notable impact on soil properties. These organisms decrease the toxicity of the soil, enhance the resistance of plants to stressors, and stimulate plant growth [160]. Microbial fertilizers not only drastically reduce the application of chemical fertilizers but also minimize the utilization of pesticides [161]. Reports have shown that the use of resilient plant growth-promoting inter-root microorganisms can be used to convert chemical pesticides into non-toxic chemicals through detoxification, degradation, complex formation and activation using the microorganisms’ own mechanism of action [162]. Finally, Table 5 summarizes the examples of the microbial remediation of soil pollutants in recent years. 

## 5. Future Perspective and Challenges

Microbial fertilizers are currently being used as an effective method of increasing crop yields. On one hand, they improve the nutrients necessary for plant growth and development. On the other hand, they are environmentally friendly to the soil and protect plants from environmental stresses. In recent years, biofertilizer has emerged as a green and sustainable development strategy that contributes to the future agricultural ecosystem’s sustainability. Additionally, the accurate application of microbial fertilizers should include a consideration of the interaction between soil properties, the soil environment, and host plants of strains to ensure precise biological fertilizer application. This approach helps minimize the impact on the agricultural ecosystem and enhance its efficiency. Overall, microbial fertilizers have a wide range of applications and serve as a green strategy to promote the sustainable development of agriculture [170]. We should have confidence in conducting further in-depth research for future investigations.

As we said, microbial fertilizer is a promising biofertilizer; however, it has some limitations and challenges. We should focus on the main challenges in the production of microbial fertilizer in view of food security. The most important choice in microbial production is that of a suitable carrier [171,172]. Apart from traditional organic wastes like straw and feces, carriers such as domestic garbage, coal powder, and nutrient soil are now being used. However, these carriers have complex compositions, and there is a risk of heavy metal and pathogenic bacteria residues, which could harm agricultural products. Microbial fertilizers are beneficial to plant growth, but they do not fully satisfy the needs of the plant and are subject to a number of environmental conditions. In addition, microorganisms may produce other substances in the process of degrading soil pollutants, which may disturb the ecological balance. Therefore, further research is necessary to understand microbial fertilizer in this context and enable its large-scale application and commercialization. In order to overcome these challenges, researchers should continue to explore several related strategies to achieve sustainable agricultural development and solve food security problems.

## 6. Conclusions

Despite the escalating state of soil environmental degradation and limited cropland, the growing population necessitates an augmentation of food production. Chemical fertilizers significantly augment crop yield in a short time span. To circumvent the adverse effects of chemicals, eco-friendly substitutes have been identified. The utilization of microbial fertilizers synthesized from PGPR is an ecologically sustainable agricultural strategy. By establishing a mutually beneficial interaction model among host plants, PGPR, and soil, these interactions regulate plant growth, resist environmental stress, rehabilitate contaminated soil (Figure 1). In summation, an overwhelming reliance on chemical fertilizers perpetuates ecological imbalance. Biological fertilizer composed of beneficial PGPR strains possesses numerous advantages. It is cost-effective, has significant potential for plant growth enhancement, improves plant resilience, and serves as a pivotal strategy for sustainable green agricultural development.

## Figures and Tables

**Figure 1 plants-13-00346-f001:**
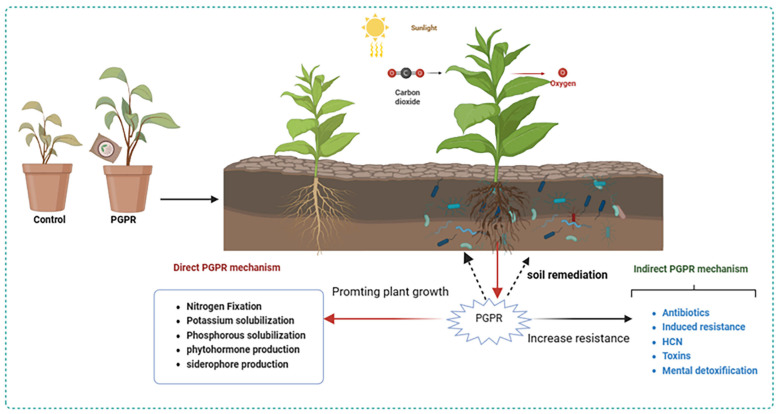
Schematic representation of the mechanisms of microbial fertilizers on crop growth and soil regulation. (Image was created with the tools of Biorender (https://app.biorender.com/, accessed on 18 November 2023).

**Table 2 plants-13-00346-t002:** Mechanisms of crop growth regulation by various beneficial microbes.

Microorganisms	Crop	Mechanism of Action	References
*Pseudomonas* sp.	Maize, cassava, spring wheat, tomato, Arabidopsisthaliana	Promotion of nutrient uptake, regulation of hormone levels, ISR, ACC-deaminase activity, siderophore, nitrogen fixation, solubilization of phosphorus	[41,42]
*Bacillus* sp.	Soya, oriental melons, potatoes, barley, maize	Vocs, antibacterial compound, organic acids, exopolysaccharides, different enzymes, ISR	[43,44,45]
*Rhizobium* sp.	Soya, peanuts,	Nitrogenfixation, exopolysaccharides, phosphate solubilization	[46,47]
*Azotobacter* sp.	Rice, tomato, cowpea bean,	Nitrogenfixation, dissolved phosphorus and potassium, generation of IAA, siderophore	[48,49,50]
*Azospirillum* sp.	Cucumber	Siderophore, indole-3-aceticacid, ISR	[51]
*Pseudomonasputida*	Melon	Different enzymes, solubilization of phosphorus, siderophore	[52]
*Pseudomonasaeruginosa*	Tobacco	Dissolved phosphorus and potassium, growth hormone	[53]
*Bacillusaryabhattai*	Tomato, maize, bean	Growth hormone, solubilization of phosphorus	[54]

**Table 4 plants-13-00346-t004:** Examples of abiotic stress mitigation by different species of PGPRs.

Microorganisms	Crop	Type of Abiotic Stress	Mechanism of Action	References
*Bacillus* licheniformis K11	Pepper	Drought stress	Auxin and ACC deaminase producing PGPR *B. licheniformis* K11 could reduce drought stress in drought-affected regions	[140]
*Bacillus subtilis*-FAB1, *Pseudomonas azotoformans*-FAP3	Wheat	Drought stress	FAB1 and FAP3 strains show unique multifunctional plant growth-stimulating properties and effective root and rhizosphere colonization to promote wheat growth during drought	[121]
*Phyllobacterium brassicacearum*	Arabidopsis thaliana	Drought stress	Bacteria induce growth and development and coordinate to improve water use efficiency in plants	[141]
*Bacillus subtilis*, *Bacillus pumilus*	Cotton	Salt stress	*B. subtilis* and *B. pumilus* significantly enhance salt stress tolerance in cotton plants during salt stress conditions.	[142]
*B. subtilis* CNBG-PGPR-1	Tomato	Salt stress	CNBG-PGPR-1 significantly improved the cellular homeostasis and photosynthetic efficiency of leaves and reduced ion toxicity and osmotic stress caused by salt in tomato	[89]
*Pseudomonas fluorescens*	Mustard	Salt stress	Two strains increase cell viability and reduce leaf damage and superoxide production	[143]
*Viridibacillus* sp.	Maize	Heavy mental stress	Inoculation with the strain promoted plant growth and development and alleviated the effects of stress on the plant	[144]
*Morganella morganii* strains	Arabidopsis thaliana	Heavy mental stress	PGPR can protect plants from Cd toxicity, and Cd-tolerant rhizobacterial strains can remediate heavy metal-polluted sites and improve plant growth	[145]
*Acinetobacter beijerinckii*, *Raoultella planticola*	Soybean	Heavy mental stress	PGPR strain promotes host antioxidant production and alters physiological and metabolic responses in soybean, enabling it to better cope with chromate and arsenic toxicity and grow well under stress	[139]

**Table 5 plants-13-00346-t005:** A case study of microbial remediation of contaminated and degraded soils.

Microorganisms	Main Pollutants	Repair Results	References
*Pseudomonas* spp., *Bacillus* substilis, *B. megaterium*	saline–alkaline soil0	These complex microbial agents could not only reduce the salt content and pH but also increase the organic content of the saline soil	[163]
*Klebsiella* sp.	Pyrene–nickel-contaminated soil	The pyrene degradation rate was 97.3% and 97.1% in pyrene-contaminated soil and pyrene–Ni-contaminated soil, respectively	[164]
*Pseudomonas*	pesticide-contaminated agricultural soil	*Pseudomonas* stutzeri CGMCC 22915 rapidly degraded sulfoxaflor to sulfoxaflor-amide via hydration.	[165]
*Bacillus megaterium*	Boron (B)- lead (Pb)- and cadmium (Cd)-contaminated soil	Reduced boron (B), lead (Pb) and cadmium (Cd) in soil and remediation of soil environment	[166]
*Mycolicibacterium* sp. Pb113 and *Chitinophaga* sp. Zn19	Heavy metal-contaminated soil	Two inoculants promote manzanita growth and improve soil zinc pollution remediation efficiency	[167]
plant growth-promoting rhizobacterium strain MD36	Heavy metal-contaminated soil	Strain MD36 effectively improves growth and yield of heavy metal-contaminated soils and bioremediation of HM-contaminated saline soils and water	[137]
*Stenotrophomonas maltophilia*	Cadmium (Cd)-contaminated soil	The SY-2 strain of *S. maltophilia* possesses significant metal tolerance and bioremediation potential against cadmium	[168]
*Acinetobacter oleivorans*	Soil contamination by hydrocarbons	*Acinetobacter oleivorans* S4 promoted plant growth and degraded total oil hydrocarbons in soil	[169]

## Data Availability

Data is contained within the review.

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
