# Peer review of "Progress in Microbial Fertilizer Regulation of Crop Growth and Soil Remediation Research"

_plants, 2024, doi:10.3390/plants13030346_

Round 1
Reviewer 1 Report
Comments and Suggestions for Authors
A thorough and comprehensive review, especially strong in the two sections culminated by Tables 1 and 2, respectively. I argue that you should include a Table 3, summarizing the section on hormonal roles. For your section on soil remediation, I am unsure if its brevity might be because not much research has been completed on this area of your topic--or if this section could be expanded (and concluded with a summary table?). Finally, I believe that your tables may be the most important part of your review--and that your closing figure serves as an important capstone for your work. Perhaps you could make your figure the centerpiece for your conclusions? And I do repeat, that I believe you need to completely revise your abstract--making it tell more of your actual "story" that you have assembled through this comprehensive work.

I have made several comments on the attached manuscript copy--most significant are some incomplete sentences--such sentences (sometimes lacking both a subject and a verb) make it hard for English language readers to understand your thoughts. I also suggest consistency in use of italics for Latin names, as well as consistency in citation style of references (within your text--at one point, you begin including authors' names--I believe that you should stick to simple numbering of references in your text).
Reviewer 2 Report
Comments and Suggestions for Authors
I commend the authors for the topic addressed.
I propose the authors to continue and expand the research especially regarding to solubilize phosphates from the soil with Plant Growth Promoting Bacteria (PGPR), the conditions and mechanisms of the process.
I propose the authors to continue and expand the research using a classification of microbial fertilizers in accordance with the existing legislation on fertilizing products (organic fertiliser, organo-mineral fertiliser, plant biostimulant).
However, special attention and new studies should be given regarding the possible phytotoxic effects and on food safety and security through the application of microorganisms, a fact that limited their inclusion in the existing, legislated and approved categories of fertilizing products.
Considering the socio-economic importance of the field, studies on the use of PGPR as fertilizing products should be directed, funded and carried out in the future as well.
Round 2
Reviewer 1 Report
Comments and Suggestions for Authors
The authors have made dramatic improvements in this paper, including a complete revision of their abstract, substantial textual enhancements throughout, and a dramatic improvement in their comprehensive tables. In my opinion, this is a strong review, and I believe that the tables alone make a significant contribution by collating in succinct form a great deal of information from this emerging research arena.
I have made a very modest number of editorial suggestions in the attached manuscript draft.
